# Frustrated self-assembly of non-Euclidean crystals of nanoparticles

Francesco Serafin[1], Jun Lu[2], Nicholas Kotov [2], Kai Sun [1] & Xiaoming Mao [1]✉

Self-organized complex structures in nature, e.g., viral capsids, hierarchical biopolymers, and bacterial flagella, offer efficiency, adaptability, robustness, and multi-functionality. Can we program the self-assembly of three-dimensional (3D) complex structures using simple building blocks, and reach similar or higher level of sophistication in engineered materials? Here we present an analytic theory for the self-assembly of polyhedral nanoparticles (NPs) based on their crystal structures in non-Euclidean space. We show that the unavoidable geometrical frustration of these particle shapes, combined with competing attractive and repulsive interparticle interactions, lead to controllable self-assembly of structures of complex order. Applying this theory to tetrahedral NPs, we find high-yield and enantiopure self-assembly of helicoidal ribbons, exhibiting qualitative agreement with experimental observations. We expect that this theory will offer a general framework for the self-assembly of simple polyhedral building blocks into rich complex morphologies with new material capabilities such as tunable optical activity, essential for multiple emerging technologies.

[1] Department of Physics, University of Michigan, Ann Arbor, MI, USA. [2] Department of Chemical Engineering, University of Michigan, Ann Arbor, MI, USA. ✉email: maox@umich.edu

Chemically synthesized nanoparticles (NPs) display a great diversity of polyhedral shapes[1]. Recent experiments revealed that under attractive interactions from van der Waals forces, hydrogen bonds, and coordination bonds, these NPs can form a number of assemblies with interesting structural order, high complexity, and hierarchy at the nanoscale, from helices to curved platelets, capsids, and hedgehogs[2–11]. How these simple polyhedral building blocks led to the observed complex structures remains an open fundamental question. Simulations of these systems face challenges from both the intrinsic complexity of NP–NP interactions and the rugged free-energy landscape of the high-dimensional phase space of their assembly[12,13]. Real-time-imaging techniques have only recently begun to reach the resolution to investigate the pathways of these self-assembly problems[14,15]. The answer to this question is not only important for emerging technologies stemming from the unique properties of these nanoscale assemblies, but also offers new insight into how complex hierarchical structures form in nature.

The mathematical problem of packing regular polyhedra in 3D Euclidean (flat) space provides a hint to answering this intriguing question. It is the rule rather than the exception that a generic polyhedron does not tile 3D Euclidean space[16]. Taking the tetrahedron as an example, one finds that five tetrahedra can form a pentamer with a small gap, and 20 tetrahedra "almost" form an icosahedron, again leaving small gaps (Fig. 1a). Perfect face-to-face attachment can only be enforced at the expense of elastic stress. Furthermore, realistic NPs also contain electrostatic charge, leading to repulsions that compete with attractions. These features make the self-assembly of polyhedral NPs an interesting "frustrated self-assembly" problem where both geometric frustration[17–23] and repulsion–attraction frustration come into play[24].

Despite the complexity originating from multiple frustrations, polyhedral NPs assembled into ordered structures such as helices in experiments[9]. We conjecture that this self-assembly phenomenon can be understood theoretically using crystalline structures of these polyhedra in non-Euclidean space. Although the assembly of most polyhedra is geometrically frustrated in

Euclidean 3D space, they can form non-Euclidean crystals in some ideal curved space, where gaps or overlaps are eliminated by precisely tuning the space's Gaussian curvature[16]. This can be illustrated by a familiar example in 2D. Regular pentagons cannot tile a 2D Euclidean surface. When positive Gaussian curvature is introduced into the surface, the gaps between the pentagons close while the plane turns into a sphere, and the pentagons fold into an unfrustrated non-Euclidean crystal: the regular dodecahedron (Fig. 1b). Similarly, any regular polyhedron can always tile as a non-Euclidean crystal in a 3D curved space[16,25], called a regular honeycomb, or polytope when the number of tiles is finite. These non-Euclidean crystals are characterized by perfect, 100% volume fraction packings of these polyhedra, and thus are the true thermodynamics ground states of the problem. General non-Euclidean crystals have been utilized in understanding complex structures of condensed matter from Frank–Kasper phases to metallic glasses, hard-disk packings, liquid crystalline order, nanoparticle supercrystals, and biological materials[25–41].

In this paper, we show that non-Euclidean crystals provide us with sets of "reference metrics" $\bar{\mathbf{g}}$[42] of the *stress-free* packing of these polyhedral NPs, characterizing their thermodynamic ground states (which cannot be realized in Euclidean 3D space), and thus offer a starting point to construct an energy functional of the assembled structures, the minimization of which guides us in the search for self-assembly morphologies.

The self-assembly we consider here are driven by competing attractive and repulsive interactions and the surface energy is typically low. These factors lead to arrested growth in certain directions, allowing a rich set of low-dimensional morphologies (e.g., 2D-sheets and 1D-bundles rather than 3D-bulk solids). A typical example of this type of NP self-assembly experiment has been described in ref. [9]. Thus, compared to "flattening" schemes of non-Euclidean crystals where disclinations are introduced to relieve the stress studied in refs. [25–30,32–38], we consider a different route of stress-relief where low-dimensional assemblies are free to choose their morphologies in Euclidean 3D space. The introduction of disclinations can further reduce the stress of the assemblies, and we leave that for future work.

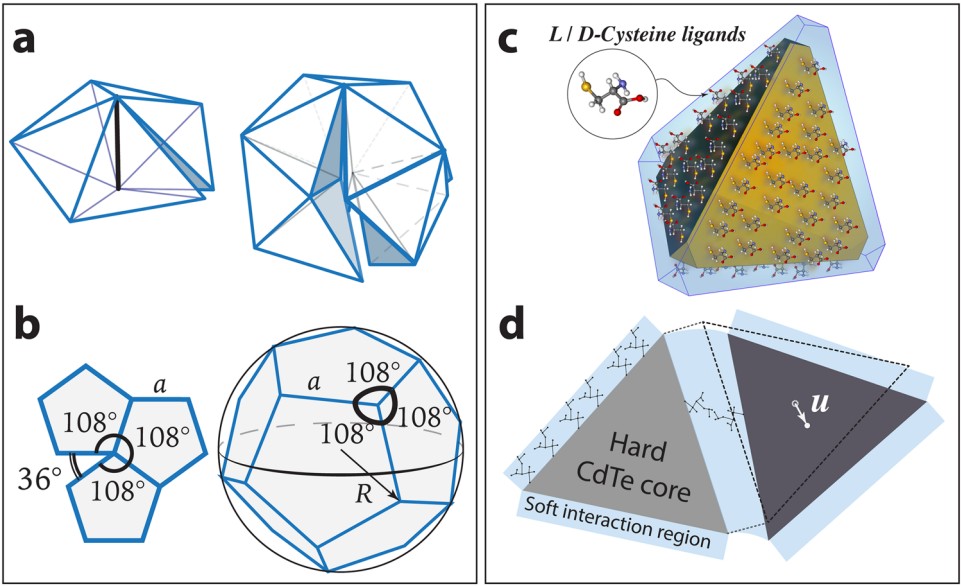

**Fig. 1 Geometric frustration of polyhedra and polygons, and tetrahedral NPs. a** Five tetrahedra can almost form a pentamer; 20 tetrahedra can almost form an icosahedron. **b** A 2D example of geometric frustration: pentagons do not tile Euclidean surface, but can form a regular crystal (dodecahedron) on the 2-sphere $S^2$. **c** CdTe tetrahedral NPs (yellow tetrahedra) are coated with a layer of chiral ligands (L- or D-Cysteine). The blue-shaded areas represent the soft interaction regions where coordination bridges between ligands can form. **d** Elastic deformations in this continuum theory represent distortions of the soft ligand interactions.

We apply our theory to this experiment where tetrahedral NPs assemble, and show good qualitative agreement between the predicted and observed trend of how the assembled morphologies depend on experimental parameters which control the interactions between the NPs. Importantly, we find that the electrostatic repulsion between the NPs provides an experimentally realistic tuning knob for the final morphology of the assembly.

This theory not only can be applied to a broad range of NP assembly systems to explain and predict the morphologies of the assembly, but also brings fundamental understandings on how a completely new design space can be opened for the nanoscale self-assembly of complex, curved, and hierarchical structures from simple polyhedral and other NPs.

## Results

**Model energy of frustrated nanoparticle self-assembly.** In this section we discuss a general model energy for frustrated NP self-assembly. We will discuss in more detail how NP interactions and kinetics determine the actual form of this energy in the next section, as we apply this general model to the system of tetrahedral NPs—the experimental system studied in ref. [9] (Fig. 1c, d).

We construct the general energy of the assembly by adding up the energy associated with (i) the aforementioned elastic frustration, $E_{elastic}$, (ii) electrostatic repulsion between the NPs, $E_{repulsion}$, (iii) the boundary of the assembly, $E_{boundary}$, and (iv) binding between the NPs, $E_{bind}$,

$$E = E_{elastic} + E_{repulsion} + E_{boundary} + E_{bind}. \quad (1)$$

Although this self-assembly problem is discrete in nature as the building blocks are individual polyhedral NPs, we consider this energy in the continuum limit, where analytic results can be obtained, by modeling the NPs and coordination bonds between them as a homogeneous continuum (see more details in the section "Tetrahedral nanoparticles and their curved crystals").

In this continuum theory, the elastic energy is $E_{elastic} = \frac{1}{2} \int_{\bar{M}} \mathscr{E}_{elastic} \sqrt{\det \bar{\mathbf{g}}} \, d^3x$, where $\sqrt{\det \bar{\mathbf{g}}} \, d^3x$ is the reference volume element[43] and $\bar{M}$ is a region in the non-Euclidean crystal. The (Euclidean) actual metric of the assembly $\mathbf{g}$ cannot be equal to the metric of the non-Euclidean crystal $\bar{\mathbf{g}}$ (which describes the ideal, stress-free, distances between the NPs) everywhere, and a strain

$$\boldsymbol{\epsilon} = \frac{1}{2}(\mathbf{g} - \bar{\mathbf{g}}), \quad (2)$$

necessarily develops. Close to any local minimum, $E_{elastic}$ can be expanded in powers of the strain tensor:

$$\mathscr{E}_{elastic} = 2\mu \, \epsilon_\nu^\tau \epsilon_\tau^\nu + \lambda(\epsilon_\nu^\nu)^2, \quad (3)$$

where $\mu, \lambda$ are the Lamé coefficients, $\nu, \tau = 1, 2, 3$ are the three spatial directions and indices are contracted with $\bar{\mathbf{g}}$. These elastic constants are mainly determined by the deformation of the ligands and coordination bonds between the NPs, instead of the NPs thermselves. This $E_{elastic}$ captures the unavoidable geometric frustration of assembling polyhedra NPs in 3D Euclidean space, in a continuum limit.

The repulsion term $E_{repulsion}$ encodes screened as well as long-ranged electrostatic repulsion, commonly found in NPs. The boundary term $E_{boundary}$ describes surface energy associated with the boundary of the assembly. The binding energy $E_{bind}$ denotes the energy released while the NPs bind, and is proportional to the volume of the assembly.

As mentioned above, we are interested in the case where complex low-dimensional morphologies are adopted by the NPs to minimize the frustration in 3D Euclidean space as well as the electrostatic repulsion. This problem can be solved in two steps.

In step one, we choose the appropriate slice $\bar{M}$ from the non-Euclidean crystal, and in step two, we solve for the morphology of the assembled structure. For any given $\bar{M}$, the boundary and binding energies, $E_{boundary} + E_{bind}$, are fixed, as $E_{boundary}$ depends on the number of exposed faces of the polyhedral NPs and $E_{bind}$ depends on the volume of $\bar{M}$, both of which are fixed for a given $\bar{M}$. Thus, in the second step, morphology only depends on the combination of $E_{elastic} + E_{repulsion}$. This second step is the main theoretical advance of this paper.

The first step of determining the appropriate slice $\bar{M}$ itself is a considerably more complicated question that requires non-equilibrium statistical mechanics. The first determining factor for the choice of $\bar{M}$ comes from the kinetic pathways. Any self-assembly follows a pathway through the formation of inter-mediate structures, such as small clusters, fibers, or sheets[44,45], which are precursors to the final configuration. Correspondingly, the reference non-Euclidean crystal can often be decomposed into sub-structures representing the precursors. We associate a pathway to every decomposition of the reference non-Euclidean crystal (or equivalently, subgroups of the non-Euclidean crystal's global symmetry group). The second determining factor is the competition between all four energy terms described above. As the assembly progresses, stress builds up, often in anisotropic ways. In addition, repulsion also favors 1D and 2D assemblies. At the same time, surface and binding energy drives smooth and tight clusters. The interplay between all these effects eventually determines the slice $\bar{M}$. Similar types of problems has been studied for bundles of chiral fibers[46] and polygon assembly in 2D[20], but a general understanding for 3D self-assembly problems has not been reached yet. In the section "Tetrahedral NPs and their curved crystals" we discuss how the slice $\bar{M}$ is selected for a system of tetrahedral NPs based on kinetic pathways and energetic arguments.

**Tetrahedral NPs and their curved crystals.** Here we specialize the model to the assembly of charged tetrahedral CdTe NPs binding via chiral surface ligands. These NPs in experiments self-assemble into enantiopure and uniform helices at the scale of microns[9].

In these experiments, the NPs assemble in a mixture of water and methanol, and the surface of the NPs are coated with L- or D-Cysteine (Cys) ligands. Van der Waals forces, hydrogen bonding, and coordination bonds between ligands induce face-to-face binding of the NPs. Cadmium ions ($Cd^{++}$) are added to regulate the ligand coordination bonds between the NP's surfaces (see the "Methods" section for more details of the experiment).

We apply the general energy introduced in the section "Model energy of frustrated nanoparticle self-assembly" to this problem of frustrated assembly of tetrahedra NPs (Fig. 1c and d). The continuum approach of this theory is justified by considering the interactions between the NPs. The NPs are polydisperse with sizes in the range of 3–5 nm, while the size of the "coordination bridge" (the Cys ligands on both NPs and the $Cd^{++}$ ion in between) is about 1 nm. Thus, rather than a hard-polyhedra model, it is more appropriate to model these NPs (including their ligands) as deformable tetrahedra, and the assemblies as an elastic continuum. Note that the elasticity of this continuum mainly comes from the variation of the interaction energy of the NPs as the NPs displace and rotate relative to one another, and not the elasticity of the CdTe NP cores, which are very stiff.

A term-by-term decomposition of the model energy in Eq. (1) can be analyzed as follows for this experimental system. The term $E_{bind}$ is the binding energy from the Van der Waals, hydrogen bonding, and coordination bonds between the NPs, assuming perfect face-to-face binding. This perfect binding is geometrically

frustrated, and the energy cost of the frustration is modeled as $E_{\text{elastic}}$, where the elastic constants $(\mu, \lambda)$ describes how the binding energy varies as the NP-NP attachment deviate from the perfect bonding. The electrostatic repulsion of the charged NPs determine the $E_{\text{repulsion}}$ part. The surface term $E_{\text{surface}}$ is the energy cost associated with the surface between the assembly and the solution.

As mentioned in the section "Model energy of frustrated nanoparticle self-assembly ", $E_{\text{bind}} + E_{\text{surface}}$ depends on how the slice $\bar{M}$ is cut from the non-Euclidean crystal, and is not affected by the morphology. On the other hand, $E_{\text{elastic}} + E_{\text{repulsion}}$ selects the morphology after the cut $\bar{M}$ is given. In this paper, we focus our quantitative theory on determining the morphology using $E_{\text{elastic}} + E_{\text{repulsion}}$ for a given region $\bar{M}$ that is a sheet from the non-Euclidean crystal. We include qualitative discussions on general principles on what determines the cut of $\bar{M}$ regarding both the competition between $E_{\text{elastic}} + E_{\text{repulsion}}$ and $E_{\text{bind}} + E_{\text{surface}}$ and kinetic pathways in the sections "Thin shell self-assembly" and "Helicoidal morphology of NPs assemblies and comparison with experiments".

The essential step to construct $E_{\text{elastic}}$ is to find the ideal metric $\bar{\mathbf{g}}$, which describes the stress-free distance between the tetrahedra, which is inherited from the non-Euclidean crystal. For these tetrahedra NPs, we choose to start from the 600-cell polytope, which is a periodic tiling of the 3-sphere $S^3$ (i.e. the surface of a ball in 4D Euclidean space) by regular tetrahedra with the lowest curvature, and thus least stressed in Euclidean space.

To understand the structure of the 600-cell (detailed in SI section I), let us start by considering how tetrahedra assemble when they are brought together by attractive interactions. It is well-known that they can form infinite straight 1D helices with no frustration, either left-handed (LH) or right-handed (RH), called "tetrahelices" or Bernal spirals (Fig. 2a)[47].

The chiral Cys ligands induce a small rotation angle between two bound tetrahedral NPs, rather than perfect face-to-face binding. As shown in Fig. 2b, this twist energetically selects tetrahelices with the same handedness as the ligands. Under the spontaneous tendency of tetrahedra to form tetrahelices, even a small chiral bias in the ligands can propagate along the tetrahelix, giving it the same handedness. In the following discussion, we only use this chiral symmetry breaking to select the fibration, but geometrically we still use the undistorted 600-cell as the reference metric. The twist due to the ligands is a perturbative effect that can be ignored in this initial consideration.

Self-assembly of tetrahelices is geometrically frustrated in 3D Euclidean space, because the twist forbids two homochiral tetrahelices to be glued side-to-side (Fig. 2c), but it can be realized on the hypersphere $S^3(R)$ of radius $R$ in Euclidean 4D space, where twist is compensated by curvature. The 3-sphere's radius $R$ is fixed by the tetrahedron's size $a$: $R = \phi a$ with $\phi$ being the golden ratio $\phi = (1 + \sqrt{5})/2$. The tetrahelices appear as closed parallel rings of 30 tetrahedra touching perfectly side to side (Fig. 2d). 20 such (homochiral) terahelices organized in four nested toroidal shells form the 600-cell regular polytope (Fig. 2f±i) which is a regular tiling of $S^3$ with tetrahedra[48]. The global structure of linked rings has the topology of the Hopf fibration[49,50] (Fig. 2e). Starting from all RH or all LH tetrahelices, this construction leads to the same (achiral) 600-cell, so the latter has two chiral decompositions: one contains all LH-, and the other all RH-tetrahelices. The 600-cell has the lowest curvature among regular polytopes formed by tetrahedra (hence a relatively low frustration) so we take it as the reference configuration for these self-assembling NPs.

It is convenient to parameterize the ideal packing of the tetrahelices in the 600-cell with angular coordinates $\Phi^\mu = (\alpha, \beta, \theta)$

on $S^3$, where the $\alpha$-axis is orthogonal to the concentric toroidal shells, the $\theta$-axis is aligned with the vertices of RH-tetrahelices, and the $\beta$-axis is aligned with the vertices of the LH-tetrahelices (Fig. 2j and SI section I). The reference metric $\bar{\mathbf{g}}$, which describes the ideal, stress-free distances of the tetrahedra packing, takes the following form in the coordinates $(\alpha, \beta, \theta)$

$$\bar{g}_{\mu\nu} = \begin{pmatrix} 1 & 0 & 0 \\ 0 & 1 & -\cos 2\alpha \\ 0 & -\cos 2\alpha & 1 \end{pmatrix}. \qquad (4)$$

The metric depends on $\alpha$ only, so the surfaces $\Sigma_\alpha$ ($\alpha = \text{const.}$) are flat tori, as shown by Bianchi in 1894[51]. More detailed discussions of this reference metric can be found in SI section II.

**Thin shell self-assembly.** The next step is to choose the slice $\bar{M}$ which represents the low dimensional assembly (with the thickness $h$ much smaller than width $W$ and length $L$) at low surface tension. In this paper we choose to study the (one-tetrahedron-thick) shell between two special toroidal surfaces $\Sigma_{\text{N,S}}$ located at $\alpha = \alpha_{\text{N,S}} \equiv \frac{\pi}{4} \mp \arctan\frac{1}{2}$ (with the mid-surface at the Clifford torus $\alpha = \pi/4$) as our slice $\bar{M}$ (Fig. 2k and l).

This choice of $\bar{M}$ is justified as follows. First, from the kinetic pathway perspective, evidences from TEM images taken at different stages of the self-assembly process indicate that tetrahelices form first and they later combine into the micro-sized helical ribbons as their final assembly[9]. This leads to a natural stress-free direction which is along the tetrahelices from the early stage of the assembly. The growth of the assembly along this stress-free direction is only limited by the non-equilibrium nucleation process of the assembly, and not by stress, so it can reach the scale of microns, as observed in the experiment. In particular, as shown in Fig. 2l, tetrahedra coated with L-Cys form LH tetrahelices, leading to a low-stress direction along $\beta$, and tetrahedra coated with D-Cys form RH tetrahelices, leading to a low-stress direction along $\theta$. Second, from the energetic perspective, as the tetrahelices bind along directions perpendicular to this low-stress direction, repulsion favors the growth of thin sheets rather than thick bundles, and the stress from geometric frustration limits the width of the sheet. Furthermore, surface and binding energies favor smooth and tight clusters.

The choice of the shell between two special toroidal surfaces $\Sigma_{\text{N,S}}$ satisfies these considerations at the same time. It contains both the LH and RH tetrahelices, which are the stress-free directions of NPs with L-Cys and D-Cys ligands, respectively. Top and bottom surfaces of this shell are both smooth triangulated surfaces, giving low surface energy. Next, we use Eq. (4) in Eq. (1) to compute the effective energy of the thin assembly, and minimize it to find the morphology of the assembly.

Since tetrahelices in 3D Euclidean space are open chains and not closed rings, we should interpret the 600-cell order only as a local reference configuration: the global topology of the tetrahelices is not fixed by the topology of the reference 600-cell. This is different from e.g. models of the cholesteric blue phase, which used the global topology of the 3-sphere[33] to find the frustrated ground state. Therefore, we cut the shell along the $\theta$ and $\beta$ directions into an open rectangular prism (where $\theta, \beta$ are no longer bounded by $[0, 2\pi]$, Fig. 2l). This also justifies our approximation of this sheet as an elastic thin sheet, as the width and length can grow much greater than the thickness (which the size of one tetrahedron).

We expand $\bar{g}_{ij}(\alpha)$ around the mid-surface

$$\bar{g}_{ij}(t) = \bar{a}_{ij} - 2t\bar{b}_{ij} + t^2\bar{c}_{ij} + o(t^3), \qquad (5)$$

where we defined the parameter $t \equiv R(\alpha - \pi/4)$ along the thickness direction, similarly to a thin shell in elasticity[43,52]. Directly from

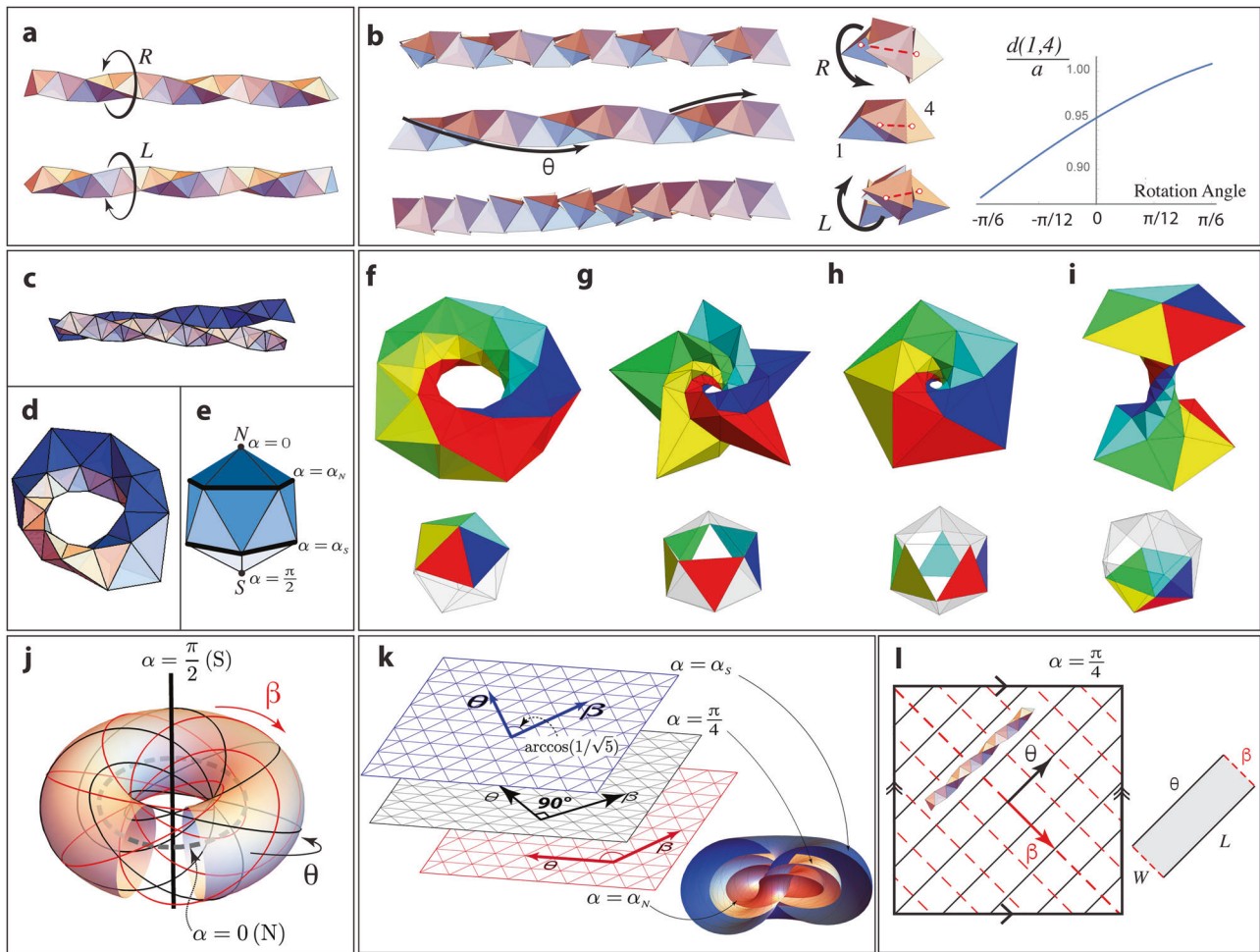

**Fig. 2 Structure of the 600 cell. a** Tetrahelices are chiral linear assemblies of tetrahedra. **b** Chiral ligands induce a twist between neighboring tetrahedra, and select and handedness of the tetrahelix. When the tetrahelix and the relative twist have the same (opposite) handedness, the distance between 4th-nearest neighbor tetrahedra's centers $d(1, 4)$ increases (reduces), thus lowering (increasing) their electrostatic repulsion. At the same time, the edges of the tetrahelix become more (less) coiled. **c** In 3D Euclidean space, two tetrahelices cannot fit side-to-side. **d** Two tetrahelices fitting perfectly together on the 3-sphere $S^3$ (a stereographic projection). **e** The base icosahedron of the 600 cell. The 4 toroidal shells are indicated by color gradient, with the $\alpha$ coordinate of the boundaries between them marked. **f–g** 4 toroidal shells of the 600 cell, each containing 5 tetrahelices. **f** North pole bundle ($0 < \alpha < \alpha_N$): 5 tetrahelices around the North pole of the base. **g** North ring of 5 tetrahelices wrapped around the North pole bundle (the latter is not visible). **h** South ring of 5 tetrahelices wrapped around the North ring. The surface triangles are facing the South pole of the base. The north and south rings together occupy the space $\alpha_N < \alpha < \alpha_S$. **i** South pole bundle ($\alpha_S < \alpha < \pi/2$): 5 tetrahelices are closely packed around the South pole of the base. **j** The coordinate $\Phi^\mu = (\alpha, \beta, \theta)$ (stereographic projection). **k** The slice $\bar{M}$, which contains the north and south rings (**g** and **h**, a total of 300 tetrahedra arranged in 10 tetrahelices, corresponding to the middle 10 triangles in the base icosahedron in **e**), are characterized by layers of constant $\alpha$ surfaces within $\alpha_N < \alpha < \alpha_S$. The angle between $(\theta, \beta)$ evolves across the layers, according to the reference metric [Eq. (4)]. **l** The middle layer of $\bar{M}$ is the Clifford torus ($\alpha = \pi/4$, represented as a square with opposite edges identified). The $\theta$ axis is along the RH tetrahelices, and along the long side of the ribbon when RH tetrahelices assemble. The assembled ribbon (gray rectangle) has reference metric $\bar{g}$ from the 600 cell but is not limited by the size of the Clifford torus.

the 600-cell, the shell we consider here includes the north and south rings (Fig. 2g and h), $\alpha_N < \alpha < \alpha_S$, which leads to a total thickness of the shell $h = 2\arctan(1/2)R$. In order for the thin shell approximation to be valid, we need the thickness to be much smaller than both the width and the length, as well as the radius of curvature. As we discuss below when comparing the theoretical and experimental results (see the section "Helicoidal morphology of NPs assemblies and comparison with experiments"), this criterion is indeed satisfied in the cases we explore here. The reference first and second fundamental forms $\bar{a}_{ij} \equiv \bar{g}_{ij}(0)$ and $\bar{b}_{ij} \equiv -\frac{1}{2R}\partial_t\bar{g}_{ij}(0)$ are

$$\bar{a} = \begin{pmatrix} 1 & 0 \\ 0 & 1 \end{pmatrix} \quad \text{and} \quad \bar{b} = \frac{1}{R}\begin{pmatrix} 0 & -1 \\ -1 & 0 \end{pmatrix}. \quad (6)$$

The reference first fundamental form $\bar{a}$ represents the ideal in-

plane metric of the mid-surface. The reference second fundamental form $\bar{b}$ represents the reference curvature. It is off-diagonal, so it favors pure twist around the axis defined by the tetrahelix direction. The twist between the contact surfaces of the tetrahelices packed in the shell generates stress between the upper and lower surfaces of the sheet, as depicted in Fig. 2k. This geometric frustration manifests in the fact that $\bar{a}$ and $\bar{b}$ are incompatible in a surface embedded in Euclidean space for which $\det\bar{b}/\det\bar{a} = -1/R^2$, while $K(\bar{a}) = 0$, violating Gauss' Theorema Egregium $\det\bar{b}/\det\bar{a} = K(\bar{a})$. Note that the Gaussi–Codazzi–Peterson–Mainardi (GCPM) equations are satisfied.

We minimize the energy of the shell to find the actual first and second fundamental forms $a$ and $b$. We first consider the

elasticity part,

$$E_{\text{elastic}}^{\text{shell}} \simeq E_{\text{elastic}}^{\text{stretch}} + E_{\text{elastic}}^{\text{bend}} = \int_{\bar{M}_0} dA \, [\mathscr{e}_{\text{elastic}}^{\text{stretch}} + \mathscr{e}_{\text{elastic}}^{\text{bend}}] \quad (7)$$

where now $\bar{M}_0$ is the Clifford torus and $dA = R^2\sqrt{\det \bar{\mathbf{a}}} \, d^2\Phi$ is the area element of the mid-surface. The out-of-plane bending energy density $\mathscr{e}_{\text{elastic}}^{\text{bend}}$ depends on $\mathbf{b} - \bar{\mathbf{b}}$

$$\mathscr{e}_{\text{elastic}}^{\text{bend}} = \frac{\kappa}{2}[(1-\nu)\text{Tr}(\mathbf{b} - \bar{\mathbf{b}})^2 + \nu\text{Tr}^2(\mathbf{b} - \bar{\mathbf{b}})] \quad (8)$$

and the in-plane stretching energy, $\mathscr{e}_{\text{elastic}}^{\text{stretch}}$ depends similarly on $\mathbf{a} - \bar{\mathbf{a}}$ with $\kappa$ replaced by $k$, with stiffness $k \equiv \frac{hY}{8(1-\nu^2)}$ and $\kappa \equiv \frac{h^3 Y}{12(1-\nu^2)}$, where $Y, \nu$ are the Young's modulus and Poisson ratio. It is worth pointing out that although the thickness scaling of the stretching and bending elastic energies are derived in the continuum limit, they represent a good approximation for an assembly of rigid NPs connected by soft ligands, as long as the curvature is not too large and a continuous limit of the deformation field is well-defined. The agreement with experimental results verified this continuum approach. A more detailed derivation of this thin shell elastic energy can be found in SI section III.

This mode of geometric frustration is similar to a number of problems in the literature on elasticity of twisted ribbons[53–57]. The minimization of this type of elastic energy $E_{\text{elastic}}^{\text{shell}}$ has analytic solutions in two limits[55–57]: "wide" ribbons ($W \gg \sqrt{Rh}$) are stretching-dominated, so $\mathbf{a} \simeq \bar{\mathbf{a}}$, and the solution is a cylindrical helical ribbon. "Narrow" ribbons ($W \ll \sqrt{Rh}$) are bending-dominated, so $\mathbf{b} \simeq \bar{\mathbf{b}}$, and the solution is a helicoid. This crossover comes from the competition between the bending energy, $E_{\text{bend}} \sim LW^5\kappa^4$, and the stretching energy, $E_{\text{stretch}} \sim h^2 \cdot LW\kappa^2$ (see ref. [55]).

Interestingly, the assembled ribbons observed in ref. [9] and experiments we perform in this paper have $W > \sqrt{Rh}$, which may lead to the conclusion that they belong to the wide ribbon limit. However, the observed morphologies are much closer to helicoids. As we analyze below, this is due to the bending stiffening effect of the electrostatic repulsion.

**Bending stiffening from electrostatic repulsion.** In all realistic cases, NPs carry some charge from spontaneous ionization of their surface and adsorptions of charged species from the media. Tetrahedral CdTe NPs in this experiment are negatively charged. The electrostatic repulsion, screened by ions in the solution, effectively stiffens the bending rigidity and enlarges the bending-dominated regime to much wider ribbons. Assuming a uniformly charged shell with total charge $q$ and charge density $\rho = q/hWL$, the potential at a point $\mathbf{R}(\sigma)$ on the sheet is

$$\phi(\sigma) = \frac{h\rho}{4\pi\epsilon} \int_{\bar{M}_0} dA(\sigma') \, \frac{1}{d(\sigma,\sigma')} \exp\left(\frac{-d(\sigma,\sigma')}{\xi}\right) \quad (9)$$

where $\sigma$ is the coordinate of the 2D sheet, $d(\sigma,\sigma') = |\mathbf{R}(\sigma) - \mathbf{R}(\sigma')|$ is the 3D Euclidean distance between two points on the sheet, $\epsilon$ is the dielectric constant, and $\xi$ is the Debye screening length, which depends on the solvent. We will study the regime $h \ll \xi \ll W$, where repulsion has a significant effect on the bending stiffness, but is still a short range force compared to the width and length of the sheet.

The electrostatic energy density $\mathscr{e}_{\text{rep}} = h\rho\,\phi(\sigma)$ can be written as an effective bending energy (SI section IV):

$$\mathscr{e}_{\text{repulsion}} = \frac{\pi}{8}\frac{h^2\rho^2\xi^3}{4\pi\epsilon}\left[2\text{Tr}(\mathbf{b}^2) - (\text{Tr}\mathbf{b})^2\right]. \quad (10)$$

We neglected corrections to the stretching energy because in the

thin sheet limit, the stretching and bending elastic energies are $\mathcal{O}(h)$ and $\mathcal{O}(h^3)$ respectively, whereas Eq. (10) is $\mathcal{O}(h^2)$. Summing $\mathscr{e}_{\text{repulsion}}$ with $\mathscr{e}_{\text{elastic}}^{\text{bend}}$ [Eq. (8)] gives the effective bending energy

$$\mathscr{e}_{\text{eff}}^{\text{bend}} = \frac{\kappa_{\text{eff}}}{2}[(1-\nu_{\text{eff}})\text{Tr}(\mathbf{b} - \bar{\mathbf{b}}_{\text{eff}})^2 + \nu_{\text{eff}} \, \text{Tr}^2(\mathbf{b} - \bar{\mathbf{b}}_{\text{eff}})], \quad (11)$$

where

$$\kappa_{\text{eff}} = \kappa + 2Q, \quad (12)$$

$$\nu_{\text{eff}} = \frac{\kappa\nu - 2Q}{\kappa + 2Q}, \quad (13)$$

and

$$Q \equiv \frac{\pi}{8}\frac{h^2\rho^2\xi^3}{4\pi\epsilon} \quad (14)$$

is the electric self-energy of a patch of size $\xi$ on the ribbon. Similar corrections to the elastic moduli were studied for charged fluid membranes in ionic solutions in refs. [58–61]. The correction to $\bar{\mathbf{b}}$ affects its traceless part, $\bar{b}_{ij}^0 = \bar{b}_{ij} - \text{Tr}(\bar{\mathbf{b}})\delta_{ij}/2$, which obtains an overall factor $\ell^{-1}$

$$\bar{\mathbf{b}}_{\text{eff}}^0 = \frac{\bar{\mathbf{b}}^0}{\ell}, \quad \ell \propto \frac{\rho^2\xi^3}{h\epsilon \, Y} \quad (15)$$

up to numerical factors of order 1, while the trace part remains 0.

Thus, the repulsion has two effects: it increases the bending rigidity [Eq. (12)] and lowers the the curvature of the reference metric [Eq. (15)]. These two effects are related, as $Q/\kappa \sim \ell$, so in the limit of strong repulsion, $\ell \gg 1$ and bending rigidity is dominated by repulsion.

An important consequence of the correction is that the reference radius $R$ of $S^3$ is enlarged into $\ell R$. In this strong repulsion regime, the characteristic length scale for the bend-stretch crossover, $\sqrt{\ell Rh}$, can exceed the physical width ($W \ll \sqrt{\ell Rh}$), bringing a ribbon into the bending-dominated regime even when $W > \sqrt{Rh}$. The morphology of this regime is solved in the section "Helicoidal morphology of NPs assemblies and comparison with experiments". Using Eq. (15) in $W \ll \sqrt{\ell Rh}$, we find that the critical volumetric charge density $\rho$ above which the self-assembly is bending-dominated is

$$\rho > \rho_c = \frac{2}{3}\left(Y\frac{\varepsilon\varepsilon_0}{\xi^3}\frac{W^2 - \phi ah}{\phi a}\right)^{\frac{1}{2}}, \quad (16)$$

where $\varepsilon$ is the relative permittivity ($\varepsilon \sim 32$ for methanol at 298 K) and $\varepsilon_0$ is the vacuum permittivity.

**Helicoidal morphology of NPs assemblies and comparison with experiments.** Without loss of generality, we describe an assembly of RH tetrahelices, with long axis aligned with $\theta$. The treatment of LH tetrahelices can be generated with mirror symmetry, as we discuss below.

In the reference configuration, the RH helices are packed side by side across the direction $\beta$. The ligand's D-chirality favors the formation of long RH tetrahelices while inter-helices bonds are more frustrated, so we conjecture that the longest dimension $L$ of the thin-shell is parallel to the RH-helices. We therefore cut a rectangular region out of the Clifford torus with the long side $L$ parallel to $\theta$ and the short side $W$ parallel to $\beta$ (Fig. 2l). In the limit of $L \gg W$ the actual metric $\mathbf{a}$ needs to be independent of $\theta$, so stress grows only in the width direction, minimizing the energy. In the repulsion-controlled bending-dominated regime, we impose the constraint $\mathbf{b} = \bar{\mathbf{b}}$ and find the actual 2D metric $a_{ij}$

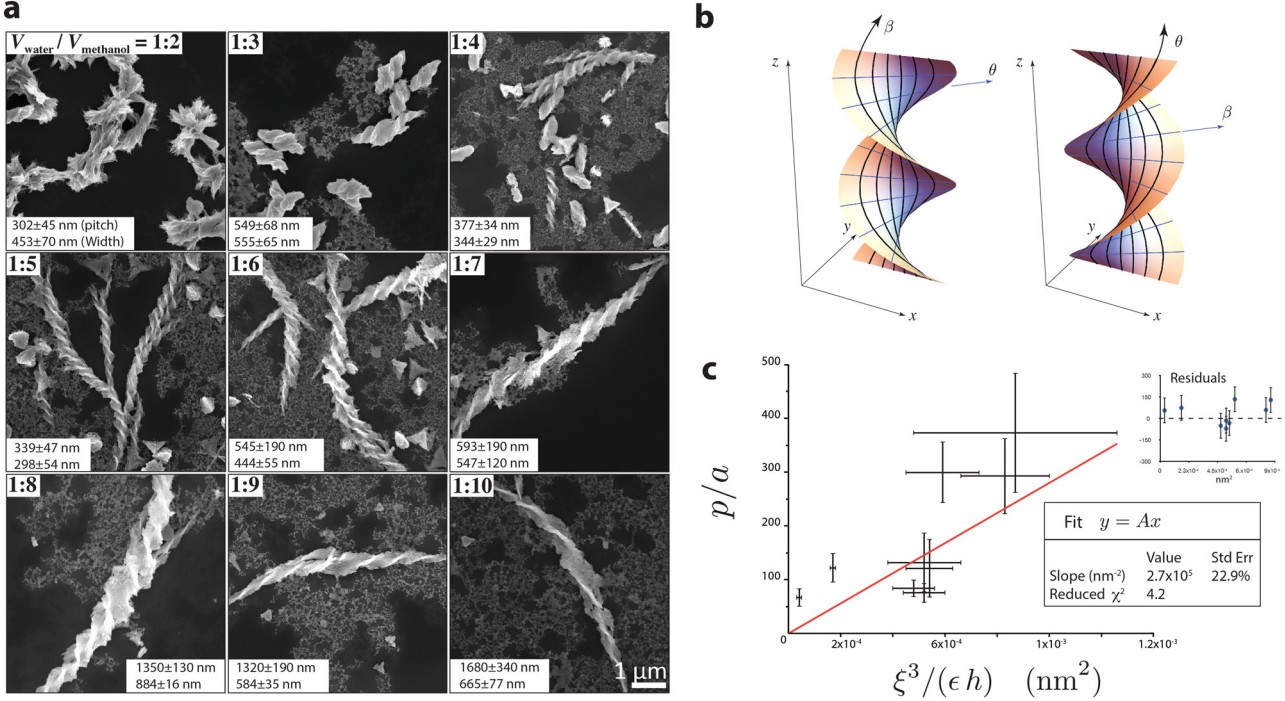

**Fig. 3 Comparing morphology of assemblies from experiment and theory. a** Experimental observation of helical assemblies of tetrahedral CdTe NPs. **b** Helicoidal solution for the ribbon's mid-surface obtained from the model. **c** Linear prediction of the helicoid's pitch on $\xi, \epsilon, h$ [Eq. (19)] (red line) fitted on the experimental data (black data points), where $a = 4.5$ nm. Error bars show the upper and lower quartiles of the distribution of pitch and thickness of the helices based on each SEM image.

by minimizing $\mathscr{E}_{\text{elastic}}^{\text{stretch}}$ (see SI section V for more details):

$$a_{ij} = \cosh^2\beta \begin{pmatrix} 1 & 0 \\ 0 & 1 \end{pmatrix}. \tag{17}$$

The embedding of the mid-surface, under free boundary conditions, is an RH helicoid (Fig. 3b)

$$\mathbf{R}(\beta, \theta) = \ell R \left( \sinh\beta\cos\theta, \sinh\beta\sin\theta, \theta \right), \tag{18}$$

where $R = \phi a$ is the radius of the 3-sphere, and $a$ is the edge length of the tetrahedra. Importantly, this RH helicoid has the same chirality as the tetrahelices and the ligands. This prediction is consistent with the structures seen in ref. [9], where high R ligand (D Cys) concentration systematically leads to RH assemblies with nearly perfect enantioselectivity. Similarly, for L ligands (L Cys), we exchange $\beta \leftrightarrow \theta$ which reverses the handedness of the tetrahelices, but leaves $\bar{\mathbf{b}}$ invariant. Imposing $\partial_\beta a_{ij} = 0$ and minimizing $E_{\text{stretch}}$, we find that now the helicoidal solution must be LH (Fig. 3b).

The pitch of the helicoid is given by

$$p = 2\pi\ell R = 2\pi \left( \frac{\rho^2 \xi^3}{h\epsilon\, Y} \right) R. \tag{19}$$

Hence the repulsion between NPs is a crucial design parameter: the pitch can be tuned by the charge density $\rho$ on the NPs, the screening length $\xi$ via the control of ion concentration in the solution, and the solution's dielectric constant $\varepsilon$.

It is worth noting that we started from a rectangular sheet of length $L$ and width $W$, but the resulting pitch $p$ is independent of $L$ and $W$. This is true for the repulsion-controlled bending-dominated regime we discussed above. In the intermediate regime where stretching and bending energies are comparable, $p$ may depend on $W$, as discussed for the purely elastic case in ref. [55].

In this experiment, the width of the helicoid is determined by the competition between the binding energy $E_{\text{bind}} \sim -\epsilon_{\text{bind}} \times$

$(hWL)$ and the (repulsion-corrected) elastic energy of the ribbon,

$$E_{\text{elastic+repulsion}} \sim Y \times (hWL) \times \left( \frac{W}{\ell R} \right)^4. \tag{20}$$

At equilibrium, $\partial(E_{\text{bind}} + E_{\text{elastic+repulsion}})/\partial W = 0$, giving

$$W \sim \left( \frac{\epsilon_{\text{bind}}}{Y} \right)^{\frac{1}{4}} \ell R \sim \left( \frac{\epsilon_{\text{bind}}}{Y} \right)^{\frac{1}{4}} \left( \frac{\rho^2 \xi^3}{h\epsilon\, Y} \right) R. \tag{21}$$

Because stress does not accumulate along the long axis of the helicoid, the length $L$ is controlled by the kinetic processes of the assembly.

In experiments, twisted sheets were produced in a mixture of water and methanol, while concentration of cadmium ions was used to control the kinetic rate of the assembly. As discussed in Methods, $\xi$ and $\varepsilon$ depend on the concentration of ions and the water/methanol ratio. The predicted dependence of the pitch on $\xi, h, \varepsilon$ [Eq. (19)] agrees qualitatively with the experimental data (Fig. 3a, c). Additionally, the measured thickness $h$ of the ribbons is much greater than a single tetrahedra (~5 nm in size). This indicate that the assembled helices in the experiment consist of multiple layers of stacked these single-tetrahedral-thick helicoids.

We are now in a position to reexamine the thin shell approximation we took here. The width and length of the assembled helicoids are of order $10^2$–$10^3$ nm and much greater than the thickness of the single-tetrahedron shell ($2\arctan(1/2)R \sim 5$ nm as we discuss below Eq. (5)). Even when considering multiple layers of ribbons stacked together, the thickness is still much smaller than the other two dimensions. In addition, the radius of curvature of the resulting morphology is at the scale of the pitch, which is $\sim\ell R$. Both this theoretical prediction and the observed radius of curvature are much greater than the thickness, justifying the thin shell approximation we took.

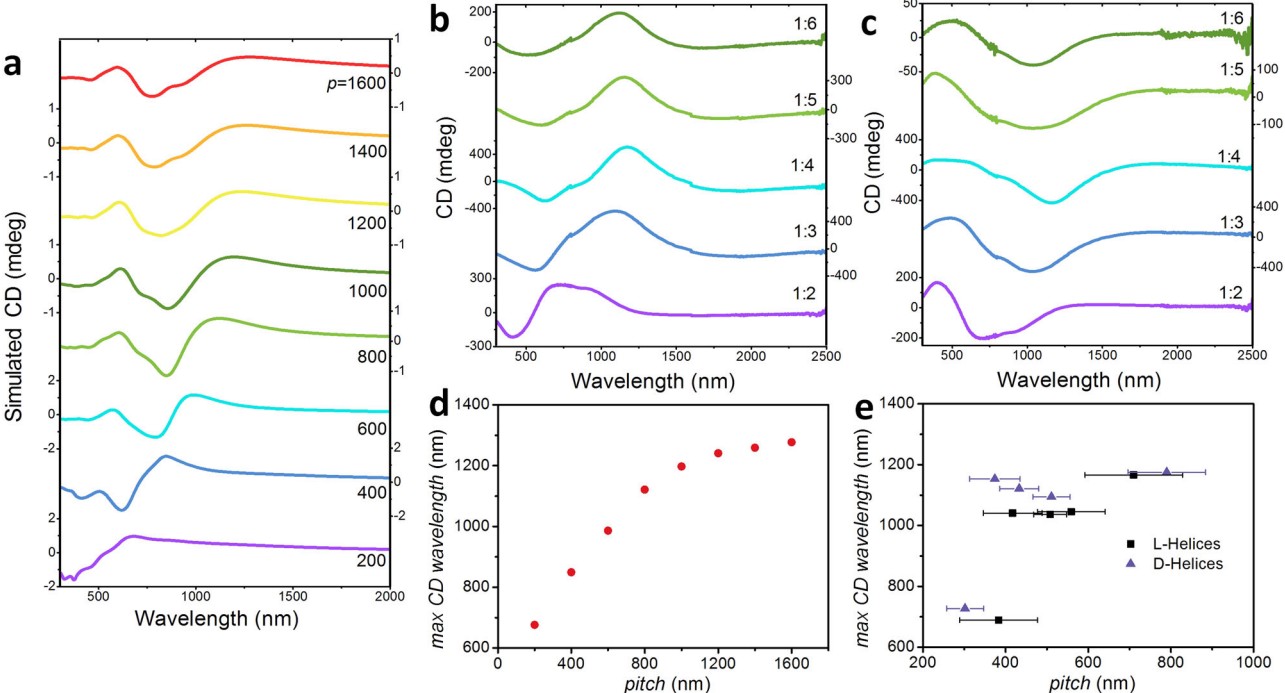

**Fig. 4 Experimental and numerical results of CD spectra of assembled structures. a** Numerical simulation of helical ribbons irradiated with circularly polarized electromagnetic waves of varying wavelength. The CD spectra are shown for helices with pitch ($p$) varied between 200 and 1600 nm. **b, c** Experimental measurement of normalized CD of [LH in (**b**) and RH in (**c**)] helices of tetrahedral CdTe NPs at water/methanol ratios 1:2–1:6 where the yield of the helices is high. Helices of longer pitch, which occur at higher methanol, are studied using simulations (**a, d**). **d, e** Characteristic CD peak wavelength plotted as function of the helical pitch, from numerical simulations (**d**) and experiments (**e**).

The theory described above provides a strategy to experimentally tune the pitch of chiral helices by adjusting charge density, solvent properties, and curvature of the reference metric, offering control of a range of physical properties. By measuring the circular dichroism (CD) spectra of self-assembled helicoids in dispersions, we find that water/methanol ratio induces different chiroptical responses at a range of wavelengths via its control of the pitch (Fig. 4b, c and see the "Methods" section). We also numerically simulated circularly polarized light (CPL) interacting with self-assembled helicoids with geometric parameters in the range produced from the experiments (varying the pitch in the range 200–1600 nm) (Fig. 4a), finding a monotonic increase of the CD peak with the pitch (Fig. 4d), the linear part (at smaller pitch) of which agreeing with experiment (Fig. 4e). Importantly, the amplitude of the CD spectra is much higher than for typical biological molecules and the maximum located in the near-infrared part of the spectrum suitable for biomedical imaging, remote sensing, and information technologies.

## Discussions

We present a non-Euclidean self-assembly theory for polyhedral NPs, which explains how complex ordered structures can be assembled from simple polyhedral NPs. We apply this theory to the geometrically frustrated self-assembly of tetrahedral NPs subject to chiral binding, and solved for helicoidal structures in agreement with experimental observations. We further show that electrostatic repulsion between the NPs provides an important tuning parameter to control the final morphology.

Although this theory focuses on the equilibrium morphology, it also provides insight into the assembly's kinetic pathways. In particular, the translational symmetry of the 2D reference metric **a̅** means that the assembly of these sheets is scalable: smaller pieces of the sheet can merge and form a larger sheet, giving the

self-assembly process a high yield of the target structure. This scalability refers to the connectivity of the assembly (topology of the contact network), instead of the morphology, which is are highly corporative ground states and depends on the size of the cluster[19]. This scalability comes from the homogeneity of the reference metric, which is translationally invariant across this sheet. As a result, how an NP connects to its neighbors is the same at different places on the sheet, allowing smaller pieces to merge. As they merge, the morphology adjusts as the cluster grows, but the topology of the contact network remains the same. If instead the in-plane metric was not translationally invariant, the local NP connectivity would be different in different locations, and the sheet could only grow from one nucleation seed, which is much slower. Interestingly, scalability is a trivial requirement for the assembly Euclidean crystals (as the metric is always flat and homogeneous), but a very nontrivial condition for cutting sub-structures from non-Euclidean crystals. The scalability of the metric greatly increases the yield, which was also observed experimentally, and provides an important measure when this theory is applied to a new new self-assembly system.

One interesting mechanism that naturally emerges in this theory is the propagation of chirality from the molecular scale (i.e., L- or D-Cys ligands on the NPs) to the assembled helices at the micron scale. As pointed out in the literature, chiral symmetry breaking mechanisms are highly nontrivial, and LH structures at the molecular scale can lead to either LH or RH structures at larger length scales, depending on the binding mechanism and the direction[62–64]. Here, the intrinsic chiral symmetry breaking of binding tetrahedra into 1D tetrahelices[65] provides a convenient channel for molecular scale chirality to propagate to the micron scale, as we discussed.

In addition, as observed in previous studies of geometrically frustrated systems, topological defects such as disclinations may arise, easing the stress at the expense of losing local attraction[66]. Here, similarly, extra tetrahelices can be inserted as disclination

lines in the 600-cell, increasing its radius $R$ and decreasing its curvature. Under this consideration, the proposed continuum model, at a lower curvature of the reference metric, can also be viewed as a continuum limit of tetrahedra assemblies with a continuous distribution of disclinations. We expect this to be a more realistic model of the experimentally observed morphologies, given their mesoscale size.

We would like to also point out interesting relations between this work and the self-assembly of amphiphilic molecules and peptides into chiral ribbons[67–69]. Although the elastic energy of these molecular assemblies shares similarities with our theory, the origin of the twist comes from chiral bonding between the molecules, which is intrinsically different from the geometrically frustrated polyhedral tiling we consider here.

This theory opens a new design space for the self-assembly of NPs, where shape and interaction of the NPs are reflected in their ideal non-Euclidean crystal structures, which in turn inform us about the self-assembly in our 3D Euclidean space. Generalization of this theory to more varieties of NPs, which exhibit tilings in either spherical or hyperbolic space, as well as diverse ways in selecting slices from these non-Euclidean crystals (e.g., clusters, tubes, shells, hierarchical structures), open a suite of intriguing new questions for future study. The new morphologies that will emerge, may lead to novel materials capabilities. Besides chiral optical response in Fig. 4, the engineering of self-assembled structures in non-Euclidean space is applicable to realization of metamaterials with unique mechanical, acoustic, catalytic, and biological properties.

## Methods

**Materials**. L-Cys hydrochloride monohydrate, hydrochloric acid (HCl) sodium hydroxide (NaOH), sulfuric acid ($H_2SO_4$, 98%) and methanol were purchased from Sigma-Aldrich. Cadmium perchlorate hexahydrate ($Cd(ClO_4)_2 \cdot 6H_2O$) was obtained from Alfa-Aesar. Aluminum telluride ($Al_2Te_3$) was purchase from Materion Advanced Chemicals. All chemicals were used as received. Ultrapure deionized water (18.2 MΩ) was used for all solution preparations.

**Synthesis of CdTe NPs**. The synthesis of CdTe NPs were according to previous publications[70] with appropriate modifications. Briefly, 0.985 g $Cd(ClO_4)_2 \cdot 6H_2O$ and 0.99 g cysteine hydrochloride monohydrate were dissolved in 100 mL deionized water. The pH of the solution was adjusted to 11.2 with 1.0 M NaOH. The obtained solution was transferred into a 250 mL three-neck round-bottomed flask and connected to a 50 mL three-neck round-bottomed flask by tubes. The system was quickly purged with nitrogen for 30 min to remove all the oxygen in the glasses and solution. Then 0.10 g $Al_2Te_3$ was added into the small flask and purged another 30 min to remove any possible oxygen in the system. 10 mL 0.50 M $H_2SO_4$ was quickly injected into the small flask to react with $Al_2Te_3$ to generate $H_2Te$ gas, which was slowly purged into the reaction solution of cadmium precursor by nitrogen flow. The reaction solution was refluxed at 100 °C for 8 h to obtain CdTe NPs in a size of 4.5 ± 0.42 nm. The as-synthesized NPs need to be wrapped with Al foil and aged as least three days before further assembly behavior.

**Self-assembly of CdTe NPs**. The self-assembly of CdTe NPs into helices with a series of pitch lengths was referred to our recent publications[9,71] with appropriated modifications. Firstly, 500 μL CdTe NPs with aging time beyond 3 days were mixed with 20 μL 0.10 M $Cd(ClO_4)_2$. The pH value of the mixed solution was adjusted to 8.0 with 1.0 M HCl. Then different volumes of methanol from 500 to 5000 μL were, respectively, added into the 500 μL above solution to initiate the self-assembly of CdTe NPs. The obtained turbid solution was incubated at room temperature under light irradiation for 3 days to assemble NPs into helices. Afterwards, the assembled helices were centrifuged at 2800 × g for 3 min and dispersed in water to wash unassembled NPs by another two times' centrifugations in the same conditions as above. The obtained helix was finally dispersed in water for further measurements and characterizations.

**Characterization**. CD and extinction spectra were acquired using J-1700 CD spectrophotometers with a PMT detector and an InGaAs NIR detector. All the spectra were measured in a quartz cuvette with a light path of 10 mm. The zeta-potential were measured by Zetasizer Nano ZSP (Malvern Instruments Ltd., GB). SEM images were taken by FEI Nova 200 Nanolab Dual Beam SEM with an acceleration voltage of 5 kV and a current of 0.4 nA. For counting the geometrical parameters of the helices, the middle region of the helices was used for analysis and more than 50 helices were counted for each case.

**Calculation of Debye screening length**. The Debye screening length, $\xi$, was calculated using

$$\xi = \sqrt{\frac{\varepsilon_r \varepsilon_0\ kT}{e^2 N_A \sum_i z_i c_i}} \qquad (22)$$

where $e$ is the elementary electric charge, $N_A$ is the Avogadro's number, $z_i$ is the charge number (valence) of $i$th component, $c_i$ is the molar concentration of $i$th component, $\varepsilon_r$ is the relative electric permittivity of the electrolyte, $\varepsilon_0$ is the electric permittivity of vacuum, $k$ is the Boltzmann constant, and $T$ is the absolute temperature. For 500 mL CdTe NPs solution before mixing with methanol, the ions in the solution were consisted of $Na^+$ (0.1635 M), $Cl^-$ (0.0851 M), $Cd^{2+}$ (0.0275 M) and $ClO_4^-$ (0.0315 M). After mixing with different volume of methanol, the concentration of each ion was diluted to different extents to get a series of Debye screening lengths.

**Dielectric constant of water/methanol mixtures**. The dielectric constants of water/methanol mixtures were according to refs. [43,72], which summarized a polynomial formula for the dielectric constant of methanol/water mixtures with the percentage of water in the mixtures based on a series of reported dielectric values:

$$\varepsilon(x) = 32.91 + 0.208\ x + 0.00246\ x^2 \qquad (23)$$

where $x$ the molar fraction of water in methanol/water mixtures.

**FDTD simulations**. The CD spectra for nanohelices with variable pitch lengths were simulated with commercial software package Lumerical FDTD Solutions. The size of nanohelices used for simulation were according to SEM images of the assembly of L-CdTe under the water/methanol ratio of 1:3, which generated a left-handed ribbon with a length, width, thickness and pitch of around 1200, 300, 100 and 600 nm. The pitch was varied from 200 to 1600 nm while kept other geometric parameters the same. To study the pitch effect on CD peak positions, the nanohelix was illuminated by left/right-handed CPL consisted by two total-field scattered-field (TFSF) sources with the same $k$-vector but with a phase difference of ±90° for left/right-handed CPL, respectively. Two analysis groups consisted of a box of power monitor were used to calculate the absorption and scattering intensity, respectively. The CD spectra were recorded as the difference of the extinction under left/right-handed CPL. The simulation wavelengths were set in the range of 300–2000 nm. The refractive index for CdTe was obtained from the *Sopra Material Database*. The refractive index of water backgrounds was 1.33. The mesh size was 10 nm. The orientation of nanohelices were considered in the simulation. The nanohelices were placed in a parallel, perpendicular and $4\pi$-averaged orientations[68,71] in comparison with the $k$-vector of incident photons, which show nanohelices under perpendicular orientation have a similar CD and extinction peak position with respect to the random orientation (see Fig. 3).

## Data availability

The data that support the findings of this study are available from the corresponding author upon reasonable request.

## Code availability

This paper presents analytic theories and fittings to experimental results, and no custom coding is involved.

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

## Acknowledgements

We thank S. Glotzer, Y. Lim, and P. Schoenhoefer for helpful discussions. This work was supported in part by the Office of Naval Research (MURI N00014-20-1-2479, J.L., N.K., K.S., and X.M.), the Department of Defense (Newton Award for Transformative Ideas during the COVID-19 Pandemic, N.K. and X.M.), National Science Foundation (NSF-EFRI-1741618, F.S., K.S., and X.M.), and Office of Naval Research for the Vannevar Bush Faculty Fellowship (N.K.).

## Author contributions

F.S., K.S., and X.M. constructed the theory. J.L. and N.K. designed and performed the experiments. All authors contributed the writing of the manuscript.

## Competing interests

The authors declare no competing interests.
