## [Peer Review File · Nature Communications]

REVIEWER COMMENTS

Reviewer #1 (Remarks to the Author):

Report on: Frustrated Self-Assembly of Non-Euclidean Crystals of Nanoparticles

Self-organisation of nanoparticles is an interesting problem which encounter several approach. Here the authors consider tetrahedral nanoparticules like CdTe and estimate energy factors occurring in such systems. It is known that tetrahedra cannot be packed in Euclidean space; this is the long story of geometrical frustration. Tetrahedra can tile a positively 3D space: the 3-sphere as the 600-polytope. Then mapping to Euclidean space needs distortions. It is the method used here.

This paper contributes interestingly to this problem and this deserve publication. Nevertheless, there is still some work to be done in order to have an easier manuscript to read for someone not specialist in geometry of curved space, in elasticity and in chemistry.

I hope that some of my remarks could help for that, but I am not a specialist of these three domains.

- The energy estimation is done in a thin shell (§IV). May be it will help to said (may be in fig2 caption) that this shell M is in S3 made of 10 Coxeter helices. So with 300 tetrahedra represented on the base of the Hopf fibration (an icosahedron, the fig 2-a) by 10 equilateral triangles.

- Two remarks about that:

- This shell in S3 is between two tori (with the Clifford torus as medium surface). It has to be flatten; all elasticity is in this operation. You consider that the shell is thin. But in S3 it contains 300 tetrahedra of the whole 600 of the polytope. It is not so thin?

- Contrarily to usual flattening methods (disclinations), you have distortions such that tetrahedra are not packed face to face (fig 1-d) you must emphasis on that.

- Do you put the S3 radius R so that NP edge length is $2\pi R/10$.

- What means "reference curvature" (in § below eq 15)?

- I have troubles with R: it is the S3 radius, the equation (eq 17, may be R is not R...) for the shell M and the right chirality, but more important R appears in eq 17 two pages after it use in page 4.

- There is nowhere reference to Van der Waals interactions which probably play a role in this case of NP.

- In §VI you consider the role of ligands. What are these ligands and their different roles? It is not clear that they impose the chirality of helices.

- Figure 2-b is not very good.

- You use the term of "low dimensional morphology" (in page 2 for instance). I suppose it means "small". But in a paper where space dimensions (3D, 4D, 2D...) appear it may be confusing.

- What means "reference curvature" (page 5)?

These are some remarks which indicate in which direction a revision of the paper is needed.

Reviewer #2 (Remarks to the Author):

This paper presents a possibly interesting approach to model the assembly of chiral helices made of tetragonal nanoparticles. Although there are potentially relevant results, I cannot recommend publication.

My main concern is that this is a difficult paper to read and evaluate its content:

- Section II describes the model energy that will be used. The problem is that it does not explain what it is that they are trying to model; the authors do not describe what is the actual experimental system that the theory aims to explain and neither what are the aspects of the experiment for which they will provide an answer.

- Then, we move to Section III, which is dedicated to the relation of the helices with the 3,3,5

polytope. For the most part, there are no new results here, but rather, it serves to fix notation and prepare the reader for the next section. For a paper in Nature Communications, there are too many words for a section that does not contain any new results.

- Section IV is where the most important results are derived. I initially was going to rederive those results, but the problem is that I could not figure out what system are the authors trying to model. Also, there are lots of approximations and although the supplementary information contains many details, it is very hard to evaluate how those affect the accuracy of the end result.

- In the end the problem is that the "trees do not allow to see the forest", with so much focus on the formalism, it is unclear how reliable are the predictions as they compare the experiments. Fig. 2d has large error bars and not that quantitative and Fig. 3 is qualitative and not very compelling.

I believe that this paper is basically two papers in one: One paper is the presentation of the formalism, where the results in supplementary information are moved to a full paper where the approximations are discussed in detail and with some (probably numerical) explaining the errors involved. Also, detailing in clear terms what is the physical system that the authors aim to model. The second paper could be the one that is eventually submitted to Nature communications, where the experimental problem is presented and compared with the actual theory, which can be presented more succinctly as the nitty-gritty are available somewhere else. I think this manuscript attempts to do too much and the focus on the formalism prevents reaching a general audience.

Reviewer #3 (Remarks to the Author):

This paper theoretically addresses the frustrated self-assembly of polyhedral nanoparticles. Based on the unfrustrated solid tetrahedra form in S^3 , solutions for the assembly process are obtained by choosing an appropriate section from the S^3 solid and seeking the least distorted immersion of it in E^3 . The work also discusses the relevance of its results to experimental observations of assemblies of tetrahedral nano-particles. The topic is timely and of wide relevance. Fabrication techniques, as well as visualization techniques, have only recently reached the level where one could ask (and also quantitatively survey) how does the local misfit of elements influences the assembled structure. Tetrahedrons are possibly the most fundamental polyhedra and are thus of exceptional importance as structural motifs. Moreover, the question of how such elements optimally pack bare relevance not only to solid crystals but to liquid crystals as well (e.g. arXiv:2103.03803 where the importance of long range tetrahedral order in 3D liquid crystals is addressed). I also found the paper well-written and exciting to read, and strongly recommend publication.

I would like to raise a few, relatively minor points for the authors to relate to at their discretion.

1. At the right column of page 2 we find $R = \phi a$, yet neither R nor ϕ were previously defined.
2. Left column page 4 : The compatibility conditions mentioned may seem synonymous with Gauss' equations whereas we know that the two Peterson-Mainardi-Codazzi equations contain independent content. Of course to show frustration it suffices to show that one of these is violated. (recently it was shown that Gauss' equation could be satisfied and the thin sheet will still show frustration due to the remaining two equations arXiv:2102.07194).
3. Tetrahedra have mirror symmetry, yet may still be considered to possess handedness (see Phys. Rev. X 4, 011003). This notion, of the underlying handedness of tetrahedra, arises in the formation of the tetrahelices as well as in the right/left equivalence of the 600 cell structure. However, I find it not sufficiently discussed in the paper. Cubes, for example, would not exhibit any chirality.
4. The bending modulus used in the paper is obtained under the assumption of continuum elasticity. This would be fitting if the primary elastic contribution would arise from local distortion of the tetrahedra. However, the majority of the elastic distortion may occur within tetrahedra facilitated by the elasticity of adjoining ligands. In this case, the assumed forms for the bending modulus and in particular, that it is cubic in the thickness may not be fully justified.
5. I find one sentence in the discussion somewhat puzzling: "the assembly of these sheets is scalable: smaller pieces of the sheet can merge and form a larger sheet"

Frustrated assemblies assume highly cooperative ground states. The optimal solutions obtained for large pieces are often very different from those obtained for small pieces. In what sense do the authors mean that the frustrated assembly is scalable?

Dear Editor and Reviewers,

We appreciate your careful consideration of our manuscript very much. Upon reviewing the referee reports we found all the questions both inspiring and helpful for us to improve the presentation of the manuscript. We believe that the deficiencies identified were all addressable and the suggestions were constructive. We made extensive changes to our manuscript to comprehensively address these issues and suggestions, which we discuss in details below. We appreciate it that these critiques and suggestions helped us making this manuscript into a better one.

In particular, we made significant additions to our discussions of the experimental system, and how interactions and kinetics in the experiment lead to our theory. We believe that the revised manuscript fully addresses concerns raised by Referees 1 and 2 on how the underlying physical mechanism is characterized in our model.

We now address points raised by the referees:

Report of Reviewer #1

Report on: Frustrated Self-Assembly of Non-Euclidean Crystals of Nanoparticles

Self-organisation of nanoparticles is an interesting problem which encounter several approach. Here the authors consider tetrahedral nanoparticules like CdTe and estimate energy factors occurring in such systems. It is known that tetrahedra cannot be packed in Euclidean space; this is the long story of geometrical frustration. Tetrahedra can tile a positively 3D space: the 3-sphere as the 600-polytope. Then mapping to Euclidean space needs distortions. It is the method used here.

This paper contributes interestingly to this problem and this deserve publication. Nevertheless, there is still some work to be done in order to have an easier manuscript to read for someone not specialist in geometry of curved space, in elasticity and in chemistry.

I hope that some of my remarks could help for that, but I am not a specialist of these three domains.

We appreciate the Referee's comment that our work "*contributes interestingly to this problem and this deserve publication.*" We have significantly revised our manuscript, adding justifications for the model, and moving some mathematical details to the Supplementary Info. We believe that it is now more straightforward for a broader audience.

- The energy estimation is done in a thin shell (§IV). May be it will help to said (may be in fig2

caption) that this shell M is in S3 made of 10 Coxeter helices. So with 300 tetrahedra represented on the base of the Hopf fibration (an icosahedron, the fig 2-a) by 10 equilateral triangles.

We appreciate this suggestion, which helps explaining the structure of the 600-cell. We have modified the caption of the new Fig 2(k) to include this point. The figures are now rearranged to better address the NP interactions in the new Fig.1.

- Two remarks about that:

- This shell in S3 is between two tori (with the Clifford torus as medium surface). It has to be flatten; all elasticity is in this operation. You consider that the shell is thin. But in S3 it contains 300 tetrahedra of the whole 600 of the polytope. It is not so thin?

- Contrarily to usual flattening methods (disclinations), you have distortions such that tetrahedra are not packed face to face (fig 1-d) you must emphasis on that.

(i) The toroidal shell we choose to study indeed contains 300 tetrahedra out of the 600 of the whole polytope. It is “thin” in the sense that it contains only one layer of tetrahedra. An important point to note here is that although the sheet is a torus in the 600-cell, the periodicity of the torus takes no effect when this sheet is assembled in Euclidean 3D space because it opens and flattens, and it can grow as big as stress permits in the L and W directions. As a result, it is a valid approximation to model them as a thin sheet in the continuum theory. We added a sentence in the paragraph above Eq.(5) “This also justifies our approximation of this sheet as an elastic “thin sheet” ...” to highlight this point.

(ii) We also added a few sentences in the introduction to highlight this point (“Thus, compared to “flattening” schemes of non-Euclidean crystals where disclinations are introduced...”, 3rd to the last paragraph in Introduction). We would like to point out that the stress-relieve scheme we adopt here is to have a low-dimensional cluster and have it adjust its morphology in the Euclidean 3D space. The chiral twist that changes the face-to-face binding in Fig.1d (the new Fig.2b) is used to explain the origin of the chirality in the assembly. We did not include this twist in the geometry—the reference metric we use is the ideal 600-cell with perfect face-to-face attachment.

- Do you put the S3 radius R so that NP edge length is 2 pi R/10.

We appreciate this clarification question, as we missed the definition of R as we edit the manuscript at submission. It is now explicitly defined that $R = \phi \cdot a$ where ϕ is the golden ratio (left hand side on page 5).

- What means “reference curvature” (in § below eq 15)?

We have modified the discussion below Eq.15 to use the more accurate description “the curvature of the reference metric” instead of “reference curvature”. We also added a sentence below Eq.6 to define \bar{b} as the “reference curvature”, following the notion used in the literature.

- I have troubles with R: it is the S3 radius, the equation (eq 17, may be R is not R...) for the shell M and the right chirality, but more important R appears in eq 17 two pages after it use in page 4.

We apologize for this confusion. We use R to denote the radius of the S3 but haven't defined it clearly. We have now added the definition $R = \phi \cdot a$ where ϕ is the golden ratio (left hand side on page 5). In the final solution for the morphology it is still the same radius, and it controls the pitch of the helicoid.

In addition, we have edited the manuscript so that we use RH and LH when we refer to chirality, to avoid confusion with the radius R.

- There is nowhere reference to Van der Waals interactions which probably play a role in this case of NP.

We thank the referee for raising this important point. The Van der Waals interactions, the hydrogen bonds, and the coordination bonds between the ligands all contribute to the short-range attraction between the NPs. These interactions control the coefficients in the model energy [Eq.(1)]. We have now extensively re-written Sec. II and III to more clearly discuss these important points.

- In §VI you consider the role of ligands. What are these ligands and their different roles? It is not clear that they impose the chirality of helices.

The ligands in the experiment are L- or D- Cysteine. They mediate an attraction between the NPs through coordination bonds with Cadmium ions. In the revised manuscript we added discussions on the experimental system in Sec III, where we clearly define the components of the experiment. Furthermore, the Cys ligands are chiral, which induces a small rotation as the tetrahedra NPs bind, and this selects the chirality of the tetrahelices and eventually the assembled ribbons. This is discussed in

Sec III (bottom left hand side). We have also added a paragraph in Conclusions to highlight this mechanism.

- Figure 2-b is not very good.

We realize that the different scales used in different panels of Fig.2b made it difficult to read. We have now changed the figure so that all the TEM images are with the same scale bar, so the morphologies at different parameters can be compared side-by-side.

- You use the term of "low dimensional morphology" (in page 2 for instance). I suppose it means "small". But in a paper where space dimensions (3D, 4D, 2D...) appear it may be confusing.

We use "low-dimensional" to mean 1D or 2D assemblies, which are "low dimensional" compared to the 3D space where the assembly is taken place, and as a result, these assemblies can adopt different morphologies via bending in different ways to relieve stress. We have added a sentence to better define this (bottom right hand side of page 1).

Report of Reviewer #2

This paper presents a possibly interesting approach to model the assembly of chiral helices made of tetragonal nanoparticles. Although there are potentially relevant results, I cannot recommend publication.

My main concern is that this is a difficult paper to read and evaluate its content:

- Section II describes the model energy that will be used. The problem is that it does not explain what it is that they are trying to model; the authors do not describe what is the actual experimental system that the theory aims to explain and neither what are the aspects of the experiment for which they will provide an answer.

We thank the referee for pointing out this problem. We have made significant additions to the revised manuscript in Sec II and III on the experimental system, and how components of the experimental system are characterized in our model. We have also added a sentence in the Introduction (end of last paragraph) pointing out that our theory can (i) explain and predict complex morphologies of existing NP self-assembly

systems, and (ii) provide a whole new design space based on the non-Euclidean crystals to program new morphologies.

- Then, we move to Section III, which is dedicated to the relation of the helices with the 3,3,5 polytope. For the most part, there are no new results here, but rather, it serves to fix notation and prepare the reader for the next section. For a paper in Nature Communications, there are too many words for a section that does not contain any new results.

We have now completely re-written Sec III, adding clear discussions of how the model characterizes various effects in the experiment, and shortened the details of the math (especially the coordinate for the 600-cell), leaving them for the Supplementary Info.

- Section IV is where the most important results are derived. I initially was going to rederive those results, but the problem is that I could not figure out what system are the authors trying to model. Also, there are lots of approximations and although the supplementary information contains many details, it is very hard to evaluate how those affect the accuracy of the end result.

We believe that with the significantly revised Sec II and III, the setup of the problem is clear, with all approximations justified. We separated the whole assembly problem into two steps, in step one a slice of the non-Euclidean crystal is selected, and in step two we calculate its morphology. Our quantitative theory is focused on the second step, but we also included careful discussions on step one (which is more challenging and needs complicated theoretical tools from non-equilibrium statistical mechanics, thus beyond the scope of this paper). The resulting morphology also agrees well with experimental observations, verifying our assumptions.

- In the end the problem is that the "trees do not allow to see the forest", with so much focus on the formalism, it is unclear how reliable are the predictions as they compare the experiments. Fig. 2d has large error bars and not that quantitative and Fig. 3 is qualitative and not very compelling.

Our intention to write this paper is to introduce a whole new theoretical framework to study frustrated NP assembly from their true thermodynamic ground states --- the non-Euclidean crystals. This is a completely new method compared to all current modeling of NP self-assembly problems. The value of this theory is to open a new design space that allows the assembly of unprecedented complex structures, instead of providing

quantitative predictions of the detailed morphologies. We have added a sentence at the end of the introduction to highlight this.

Detailed predictions of complex self-assembly morphologies of NPs is an extremely difficult problem even considering large-scale simulations, and the challenge comes from both the complexity of the NP-NP interactions, and from the out-of-equilibrium nature of the assembly process. These difficulties have been discussed in a few recent reviews, e.g., Science 350, 1242477 (2015) and Chem. Rev. 116, 11220 (2016), as well as computational studies of tetrahedral NPs specifically, J. Chem. Phys. 134, 194502 (2011), Soft Matter 15, 2260 (2019). We have added a sentence in the first paragraph of the introduction to highlight this point.

- I believe that this paper is basically two papers in one: One paper is the presentation of the formalism, where the results in supplementary information are moved to a full paper where the approximations are discussed in detail and with some (probably numerical) explaining the errors involved. Also, detailing in clear terms what is the physical system that the authors aim to model. The second paper could be the one that is eventually submitted to Nature communications, where the experimental problem is presented and compared with the actual theory, which can be presented more succinctly as the nitty-gritty are available somewhere else. I think this manuscript attempts to do too much and the focus on the formalism prevents reaching a general audience.

We are confident that the revised manuscript achieves both without obscuring the main scientific idea with too many mathematical details. As we discussed above, the merit of the paper lies in the proposal of this new theoretical framework, rather than achieving quantitative agreement with experiment, which is beyond our scope. We have followed helpful suggestions from all Reviewers, by adding clearer descriptions of the experimental system and justifications of our approximations, and moving considerable amount of details to Supplementary Info. Now the manuscript should be accessible to a broader audience.

Report of Reviewer #3

This paper theoretically addresses the frustrated self-assembly of polyhedral nanoparticles. Based on the unfrustrated solid tetrahedra form in S^3 , solutions for the assembly process are obtained by choosing an appropriate section from the S^3 solid and seeking the least distorted immersion of it in E^3 . The work also discusses the relevance of its results to experimental observations of assemblies of tetrahedral nano-particles. The topic is timely and of wide relevance. Fabrication

techniques, as well as visualization techniques, have only recently reached the level where one could ask (and also quantitatively survey) how does the local misfit of elements influences the assembled structure. Tetrahedrons are possibly the most fundamental polyhedra and are thus of exceptional importance as structural motifs. Moreover, the question of how such elements optimally pack bare relevance not only to solid crystals but to liquid crystals as well (e.g. arXiv:2103.03803 where the importance of long range tetrahedral order in 3D liquid crystals is addressed). I also found the paper well-written and exciting to read, and strongly recommend publication.

We thank the Reviewer for the positive feedback, and for pointing out the inspiring connection to tetrahedral order in liquid crystals. We have added references to this body of literature to highlight this connection in the Introduction.

I would like to raise a few, relatively minor points for the authors to relate to at their discretion.

- 1. At the right column of page 2 we find , $R=\phi a$, yet neither R nor ϕ were previously defined.*

We have added the definitions in the revised manuscript.

- 2. Left column page 4 : The compatibility conditions mentioned may seem synonymous with Gauss' equations whereas we know that the two Peterson-Mainardi-Codazzi equations contain independent content. Of course to show frustration it suffices to show that one of these is violated. (recently it was shown that Gauss' equation could be satisfied and the thin sheet will still show frustration due to the remaining two equations arXiv:2102.07194).*

This is indeed worth checking---in our case, the Peterson-Mainardi-Codazzi equations are satisfied. We have added a sentence at the end of the paragraph discussing the Gauss' equation.

- 3. Tetrahedra have mirror symmetry, yet may still be considered to possess handedness (see Phys. Rev. X 4, 011003). This notion, of the underlying handedness of tetrahedra, arises in the formation of the tetrahelices as well as in the right/left equivalence of the 600 cell structure. However, I find it not sufficiently discussed in the paper. Cubes, for example, would not exhibit any chirality.*

We thank the Reviewer for pointing out this important relation. We have now added a paragraph in Conclusions to cite the literature and highlight the mechanism of chiral symmetry breaking in this problem.

4. The bending modulus used in the paper is obtained under the assumption of continuum elasticity. This would be fitting if the primary elastic contribution would arise from local distortion of the tetrahedra. However, the majority of the elastic distortion may occur within tetrahedra facilitated by the elasticity of adjoining ligands. In this case, the assumed forms for the bending modulus and in particular, that it is cubic in the thickness may not be fully justified.

We thank the referee for raising this important point. Indeed most of the deformation is taken by the ligands, as the tetrahedra NPs are much stiffer. We believe that the h^3 scaling of bending stiffness also applies in this limit, as long as the curvature is not too large. Consider the following simplified picture of stiff squares of side length a (blue) connected by ligands (red lines) with spring constant k and rest length l , so all-squares-in-a-straight-line is the ground state. When this line is bent with curvature $1/R$, the bending angle at each junction is $\theta = a/R$, and thus the energy cost at each junction is $\int_0^h dz \frac{1}{2} k(\theta \cdot z)^2 \sim k \theta^2 h^3$, still cubic in h .

The actual geometry with tetrahedra NPs is a lot more complicated than this simple schematic, but this principle will still apply, given that the curvature is not too large. We have added a sentence in the manuscript [below Eq. (8)] to clarify this point.

5. I find one sentence in the discussion somewhat puzzling: "the assembly of these sheets is scalable: smaller pieces of the sheet can merge and form a larger sheet" Frustrated assemblies assume highly cooperative ground states. The optimal solutions obtained for large pieces are often very different from those obtained for small pieces. In what sense do the authors mean that the frustrated assembly is scalable?

By "scalable" we refer only to the connectivity of the assembly, not the specific morphology. We fully agree with the referee that the morphology the assembly takes is highly cooperative, and different system size can lead to totally different morphologies. What we tried to articulate here is that the connectivity, i.e., how many neighbors each NP connects to and which face do they attach, is *homogeneous* on this sheet we study,

given its homogeneous reference metric. Thus, as smaller pieces merge into bigger pieces, the morphology may adjust, but the assembly doesn't need to be re-connected. This might seem trivial but is often not true when cutting sub-structures from non-Euclidean crystals, as they are not characterized by simple translational symmetry as Euclidean crystals are. If one randomly choose a sheet or a ribbon from the whole non-Euclidean crystal, the metric likely will change along the sheet or ribbon, meaning that the polyhedral are connected according to different rules. In this case, the assembly is not "scalable", since when smaller pieces merge, they have to dis-assemble and re-assemble, reducing the yield of the assembly.

We have added discussions around this point (2nd paragraph in Conclusions) in the manuscript, as it is a special property for non-Euclidean crystals.

REVIEWERS' COMMENTS

Reviewer #1 (Remarks to the Author):

I think that authors have consider all my remark. So the paper can be published in it revised form.

Reviewer #2 (Remarks to the Author):

My main objection was not related to the quality or the novelty of the work, since the main result of the paper Eq. 18, and all the conceptual aspects leading to it are new and original. It was related to clarifying the derivation and the approximations leading to it, specially since the experimental evidence encoded in Fig. 3c was not absolutely compelling.

The authors have made a very significant effort to answer the referee comments and have made quite a number of changes. As a result the paper has improved a lot from its previous version and it is easier to read and follow. Given this and all the very positive comments of the other two reviewers, I recommend also the paper for publication.

Still, there are aspects that will be difficult to follow for interested readers, and I would urge the authors to make an additional effort to make them more accessible. For example:

- I do not quite follow the expansion in small powers of \hbar to obtain the metric in Eq. 5. What is the value of \hbar ? since it is used in Fig. 3c. How much "freedom" there is in this definition or how it could affect the prediction in Fig. 3c.
- I still think that it is possible to make a shorter presentation, specially in III and IV explaining the concepts behind the actual physics and relegating details to the supplementary information. The only way to read the paper right now is to go through the entire formalism, and this requires a considerable amount of time and expertise that it is not usually available to the actual reader.

Reviewer #3 (Remarks to the Author):

The authors have satisfactorily addressed all the points raised and the revised version is indeed clearer.

I strongly recommend publication.

Dear Editor and Reviewers,

Thank you very much for your careful consideration of our manuscript. We appreciate that all reviewers are satisfied with our revised manuscript, and recommend for publication. We have revised our manuscript again following the suggestion of Reviewer #2, and hereby resubmitting our manuscript for publication on Nature Communications.

Please find below our point-by-point response to questions raised by the reviewers.

Report of Reviewer #1

I think that authors have consider all my remark. So the paper can be published in it revised form.

We appreciate that Reviewer #1 found our manuscript to be ready for publication.

Report of Reviewer #2

My main objection was not related to the quality or the novelty of the work, since the main result of the paper Eq. 18, and all the conceptual aspects leading to it are new and original. It was related to clarifying the derivation and the approximations leading to it, specially since the experimental evidence encoded in Fig. 3c was not absolutely compelling.

The authors have made a very significant effort to answer the referee comments and have made quite a number of changes. As a result the paper has improved a lot from its previous version and it is easier to read and follow. Given this and all the very positive comments of the other two reviewers, I recommend also the paper for publication.

Still, there are aspects that will be difficult to follow for interested readers, and I would urge the authors to make an additional effort to make them more accessible. For example:

- I do not quite follow the expansion in small powers of h to obtain the metric in Eq. 5. What is the value of h ? since it is used in Fig. 3c. How much "freedom" there is in this definition or how it could affect the prediction in Fig. 3c.

We thank the referee for pointing out this problem, which was not completely clarified in the manuscript. In most part of the theory, we consider a single tetrahedron sheet, where the thickness at the scale of the size of one tetrahedron. In order for the thin shell

expansion to be valid, we need this thickness to be much smaller than the width and the length, as well as the radius of curvature. After we compare with experimental results, we revisited these assumptions, and verified that they are all met, even when multiple layers of the ribbon is stacked. We have included a few sentences to clarify this point in the manuscript [paragraph below Eq.(5), last paragraph of Sec. V, next-to-last paragraph of Sec. VI, and last paragraphs in Sec IV of the SI.].

- I still think that it is possible to make a shorter presentation, specially in III and IV explaining the concepts behind the actual physics and relegating details to the supplementary information. The only way to read the paper right now is to go through the entire formalism, and this requires a considerable amount of time and expertise that it is not usually available to the actual reader.---

We thank the referee for this suggestion. In the revised version we have relegated most details of the 600-cell structure to the supplementary information, and the major part of Secs. III and IV are background and justifications of the model assumptions, which we believe are necessary for the completeness of the manuscript.

Report of Reviewer #3

The authors have satisfactorily addressed all the points raised and the revised version is indeed clearer.

I strongly recommend publication.

We appreciate that Reviewer #3 found our manuscript to be ready for publication.